# ^99m^Tc-Labeled FAPI SPECT Imaging in Idiopathic Pulmonary Fibrosis: Preliminary Results

**DOI:** 10.3390/ph16101434

**Published:** 2023-10-09

**Authors:** Yu Liu, Qian Zhang, Yuwei Zhang, Jingnan Wang, Yitian Wu, Guangjie Yang, Jiyun Shi, Fan Wang, Zuojun Xu, Hongli Jing

**Affiliations:** 1Department of Nuclear Medicine, Peking Union Medical College Hospital, Chinese Academy of Medical Science and Peking Union Medical College, Beijing Key Laboratory of Molecular Targeted Diagnosis and Therapy in Nuclear Medicine, Beijing 100730, China; liuyu@pumch.cn (Y.L.); yitian_wu0@163.com (Y.W.);; 2Department of Respiratory Medicine, Peking Union Medical College Hospital, Chinese Academy of Medical Sciences and Peking Union Medical College, Beijing 100730, China; 3Medical Science Research Center (MRC), Peking Union Medical College Hospital, Chinese Academy of Medical Sciences and Peking Union Medical College, Beijing 100730, China; 4Medical Isotopes Research Center and Department of Radiation Medicine, Scchool of Basic Medical Sciences, Peking University, Beijing 100191, Chinawangfan@bjmu.edu.cn (F.W.)

**Keywords:** idiopathic pulmonary fibrosis (IPF), ^99m^Tc-HFAPI, SPECT, dosimetry

## Abstract

Aim: Idiopathic pulmonary fibrosis (IPF) is associated with a poor prognosis, presenting the most aggressive form of interstitial lung diseases (ILDs). Activated fibroblasts are crucial for pathological processes. Fibroblast activation protein (FAP) inhibitor (FAPI) tracers would be promising imaging agents for these diseases. The purpose of this study was to evaluate a ^99m^Tc-labeled FAPI tracer, ^99m^Tc-HFAPI imaging in IPF patients. Methods: Eleven IPF patients (nine males and two females; age range 55–75 year) were included in this pilot study. ^99m^Tc-HFAPI serial whole-body scintigraphy at 5 min, 20 min, 40 min, 1 h, 2 h, 3 h, 4 h, and 6 h was acquired for dynamic biodistribution and dosimetry estimation in seven representative patients. SPECT/CT tomography fusion imaging of the chest region was performed in all patients at 4 h post-injection, which was considered as the optimal acquisition time. Dosimetry was calculated using OLINDA/EXM software (version 2.0; HERMES Medical Solutions). The quantified or semi-quantified standardized uptake values (SUVs) and lesion-to-background ratios (LBRs) of affected lung parenchyma were also calculated. The high-resolution CT (HRCT) stage was determined with visual evaluation, and the total HRCT score of each patient was measured using a weighting factor formula. Pulmonary function tests (PFTs) were recorded as well. Then, the relationships between the ^99m^Tc-HFAPI results, disease extent on HRCT, and PFT results were investigated. Results: Normal physiological uptake of ^99m^Tc-HFAPI was observed mainly in the liver, intestinal tract, pancreas, gallbladder, and to a lesser extent in the spleen, kidneys, and thyroid, with no apparent retention in the blood circulation at the late time point. The mean injected activity of ^99m^Tc-HFAPI was 813.4 MBq (range 695.6–888.0 MBq). No subjective side effects were noticed. The average whole-body effective dose was 0.0041 mSv/MBq per patient. IPF patients exhibited elevated pulmonary ^99m^Tc-HFAPI uptake in abnormal lung regions, which was correlated with fibrotic regions on HRCT. Among different HRCT stage groups, both SUV_max_ and LBR showed significant differences (*p* < 0.001). The higher HRCT stage demonstrated significantly higher SUV_max_ and LBR. A linear correlation between ^99m^Tc-HFAPI uptake and total HRCT score was observed for SUV_max_ (r = 0.7839, F = 54.41, *p* = 0.0094) and LBR (r = 0.7402, F = 56.33, *p* = 0.0092). ^99m^Tc-HFAPI uptake also had moderate correlations with PFT results. Conclusions: Our preliminary data show that the ^99m^Tc-HFAPI SPECT imaging is a promising new imaging modality in IPF patients. Investigations of its clinical value in monitoring disease progression and treatment response are needed in the future.

## 1. Introduction

Interstitial lung diseases (ILDs) are a heterogeneous group of parenchymal lung diseases which can cause a highly morbid and fatal pathological response. ILDs are divided into those that are associated with known causes, such as autoimmune connective tissue diseases, drugs, radiation, and environmental agents, and those that are idiopathic, classified as idiopathic interstitial pneumonia (IIP), including idiopathic pulmonary fibrosis (IPF), non-specific interstitial pneumonia (NSIP), cryptogenic organizing pneumonia (COP), etc. [1,2]. Among them, IPF is the most common and aggressive form with an extremely poor prognosis, with a median survival of 3 years after diagnosis [3]. Other forms of ILDs display a variable clinical course. The diagnostic criteria of the American Thoracic Society/European Thoracic Society guidelines are based on clinical–radiologic distinctive features and allow a confident diagnosis in typical cases, with pathologic confirmation only in atypical cases [4].

High-resolution CT (HRCT) is the current imaging modality for IPF diagnosis and has become an integral part of the evaluation of follow-up patients. HRCT patterns, including honeycombing and traction bronchiectasis, have been reported as outcome predictors in IPF patients. However, HRCT could not assess disease activity in ILDs [5]. ^18^F-fluorodeoxyglucose (^18^F-FDG) positron emission tomography (PET) is also used for the imaging of ILDs since ^18^F-FDG uptake by tissues is a marker of glucose metabolism. It has recently been demonstrated that the ^18^F-FDG signal is raised in the fibrotic pulmonary lesions (reticulation and honeycombing) in ILDs due to increased glucose transporter-1 expression in inflammatory cells and erythrocytes, as a result of neovascularization in fibrotic areas [6]. Hence, the ^18^F-FDG signal depicts inflammatory reactions but not the activated fibrotic process.

A number of studies have highlighted the importance of the fibroblastic foci (aggregates of collagen-producing activated fibroblasts or myofibroblasts) as a manifestation of active lung injury, and their profusion may predict physiologic decline and mortality, particularly in IPF [7]. Fibroblastic foci are the key histologic feature of fibrotic progress in ILDs, and their extent correlates with a worse clinical outcome [8,9,10]. Activated fibroblasts are essential for various physiological and pathologic conditions with the remodeling of the extracellular matrix, causing progressive tissue fibrosis [11]. Fibroblast activation protein (FAP), which is overexpressed by activated fibroblasts, is associated with wound healing, fibrotic lesions, inflammation, and cancer [12,13]. Fibroblast activation protein (FAP) is highly expressed in these activated fibroblast cells. Moreover, the activated fibroblasts are essential in the pathogenesis and progression of fibrotic processes in fibrotic ILD [10].

Recently, FAP-specific inhibitor (FAPI) quinoline-based derivatives have been designed into radiopharmaceutical agents. FAPI tracers have already been developed and demonstrated promising clinical results not only in different tumor types but also in various benign diseases, including inflammatory bowel disease, arthritis, atherosclerosis and myocardial infarction, IgG4-related Disease, etc. [14,15,16]. There have been few reports of the application of FAPI tracers in ILDs. Drug-induced pulmonary fibrosis in a mouse model demonstrated that FAPIs were involved in the pathogenesis of lung tissue fibrosis. FAPI PET could detect lung fibrogenesis, both the early presence and activity of the disease, while CT was unable to assess disease activity. So, FAPI PET could be a promising tool for assessing early disease activity and evaluating the efficacy of therapeutic interventions in lung fibrosis patients [17,18]. Bergmann and Distler et al. evaluated the value of ^68^Ga-FAPI-04 PET/CT in the assessment of fibrotic activity and risk stratification in systemic sclerosis-related (SSc) interstitial lung disease [19]. Rohrich et al. found elevated ^68^Ga-FAPI uptake in fibrotic ILD lesions and lung cancer [20]. Yang et al. revealed that the FAP signal was upregulated in vitro and in vivo in ILD patients’ lungs. FAP expression level was correlated with the existence of fibroblastic foci on human lung biopsy sections from ILD patients. In addition, the total standard uptake value (SUV) of FAPI PET was significantly related to lung function decline, indicating that FAPI PET may aid in the early diagnosis and identify the perfect treatment window period of the diseases [21].

On the basis of the above findings, FAPI tracers would be promising imaging agents for ILDs. Compared to PET tracers, the single photon emission computed tomography (SPECT) tracer technetium-99 m (^99m^Tc) is more economical. ^99m^Tc-labeled FAPI imaging could be a cost-saving alternative method to PET imaging since ^99m^Tc is the most widely used radionuclide in nuclear medicine. Lindner et al. revealed that ^99m^Tc-labeled FAPI tracers showed excellent binding properties, high affinity, and significant tumor uptake in biodistribution studies in tumor-bearing mice.^99m^Tc-FAPI-34 accumulated in the tumor lesions in patients with metastasized ovarian and pancreatic cancer, as also shown on PET imaging using ^68^Ga-FAPI-46 [22]. Jia et al. developed a ^99m^Tc-labeled FAPI tracer, ^99m^Tc-HYNIC-FAPI-04 (^99m^Tc-HFAPI), which can be obtained by a reliable kit-labeling procedure [23]. They demonstrated its SPECT application in patients with digestive system tumors, and ^99m^Tc-HFAPI showed a greater diagnostic efficiency than contrast-enhanced CT. To our knowledge, the application of ^99m^Tc-HFAPI in interstitial lung diseases has not been reported yet. In this current study, we report our first clinical experience of ^99m^Tc-HFAPI in ILD patients. We aim to evaluate the distribution and uptake of ^99m^Tc-HFAPI in ILD patients. The potential role of ^99m^Tc-HFAPI in ILD patients and its relationship with pulmonary function tests and HRCT imaging are also demonstrated.

## 2. Results

### 2.1. Quality Control and Stability

Radio-ITLC was used for QC developed in ACN/Saline. As shown in Figure 1, the radiochemical purity of ^99m^Tc-HFAPI was greater than 95%. ^99m^Tc-HFAPI was stable in cystine for at least 2 h, with a radiochemical purity of greater than 95% (shown in Figure 2).

### 2.2. Baseline Demographic Data and Safety Assessment

The baseline demographic data of the study population of the 11 patients are summarized in Table 1. The eleven patients (nine males and two females, 62.0 ± 5.8 years of age) were diagnosed with IPF. All these patients were defined with the usual interstitial pneumonia (UIP) pattern according to HRCT. None of them had been treated with anti-fibrosis therapy.

^99m^Tc-HFAPI was safe and well tolerated in all subjects. With a mean injected dose of 813.4 MBq (range 695.6–888.0 MBq) of ^99m^Tc-HFAPI, no toxic-related side effects were experienced. Vital parameters remained stable, and no adverse symptoms were noticed.

### 2.3. Dosimetry and Distribution Analysis

Normal physiological uptake of ^99m^Tc-HFAPI was observed mainly in the liver, intestinal tract, pancreas, gallbladder, and to a lesser extent in the spleen, kidneys, and thyroid, with rapid clearance and no apparent retention in the blood circulation at the late time point, which was the same as in Jia et al.’s work [23]. Excessive uptake was seen in the lung area in the ILD patients at the early time points and was still visible even at 6 h after injection. A representative example of ^99m^Tc-HFAPI distribution from whole-body scintigraphy in one of the ILD patients is shown in Figure 3. In addition, one ovarian cancer patient without affected lungs was purposely enrolled to evaluate the ^99m^Tc-HFAPI distribution in the normal lung region. No abnormal uptake was observed in the healthy lung region (Appendix A).

The whole-body and organ effective doses shown in Table 2 were calculated based on the serial whole-body scintigraphy. The whole-body effective dose was 0.0041 ± 0.000559 mSv/MBq per patient, which is consistent with other ^99m^Tc-labeled agents [24].

### 2.4. 99^m^Tc-HFAPI Uptake in IPF Patients

In healthy human subjects, there is no pulmonary ^99m^Tc-HFAPI uptake (Appendix A). So, any region of tracer uptake within the lungs was considered abnormal. IPF patients exhibited raised pulmonary ^99m^Tc-HFAPI uptake, with a subpleural and peripheral distribution, involving especially the base of both lungs, as shown in Figure 4. In the corresponding areas of elevated tracer uptake, fibrotic abnormalities, including reticular opacity, traction bronchiectasis, and honeycombing, were observed in HRCT images. 

In addition, the degree of increased uptake was associated with fibrotic severity. In Figure 5, fibrotic regions were found in the subpleural of basal parts, and corresponding elevated ^99m^Tc-HFAPI uptake was observed, with an SUV_max_ of 4.5 and LBR of 3.0 in the right lung and an SUV_max_ of 4.4 and LBR of 2.9 in the left lung. A minor abnormality was also found in the apex parts, with less tracer uptake (SUV_max_ of 3.4 and LBR of 2.3 in the right lung and SUV_max_ of 3.7 and LBR of 2.5 in the left lung).

### 2.5. Relationship between ^99m^Tc-HFAPI Uptake and HRCT Findings

The severity of the affected pulmonary tissue was visually evaluated in HRCT images using a 5-point HRCT stage. Both the SUV_max_ and LBR showed significant differences (*p* < 0.001) among different HRCT stage groups (Figure 6). The mean (SD) SUV_max_ was 1.09 (0.30), 2.37 (0.77), 3.68 (0.87), 4.48 (1.16), and 3.77 (0.64), and the mean (SD) LBR was 0.74 (0.22), 1.64 (0.55), 2.47 (0.53), 2.87 (0.55), and 2.72 (0.45), for HRCT stage 0–4, respectively. The higher HRCT stages demonstrated significantly higher SUV_max_ and LBR, except for stage 4. The total SUV_max_ had a strong and statistically significant correlation with the total HRCT score (r = 0.7839, F = 54.41, *p* = 0.0094), indicating that ^99m^Tc-HFAPI uptake was comparable with disease extent on HRCT. Similarly, a significant moderate correlation between the total LBR and total HRCT score (r = 0.7402, F = 56.33, *p* = 0.0092) was observed as well (Figure 7). 

### 2.6. Correlations between ^99m^Tc-HFAPI Uptake and PFTs

Regarding the results of pulmonary tests, the total SUV_max_ had moderate correlations with CPI (r = 0.5584, *p* = 0.0742), percentage predicted DLCO (r = −0.5127, *p* = 0.1068), percentage predicted FEV1 (r = 0.−3347, *p* = 0.3144), and FEV1/FVC (r = 0.3279, *p* = 0.3249) (Figure 7). In addition, analysis of the total LBR between the above parameters also showed similarly moderate correlations.

## 3. Discussion

In our preliminary study, we report the ability of the ^99m^Tc-labeled FAPI tracer ^99m^Tc-HFAPI to evaluate fibrotic abnormalities in IPF patients. With a mean injected dose of 813.4 MBq of ^99m^Tc-HFAPI, no side effects and no adverse symptoms were noticed. The mean whole-body effective dose was 0.0041 mSv/MBq, comparable to those of known single-photon tracers [24] and lower than those of ^68^Ga- or ^18^F-labeled FAPI [25]. ^99m^Tc-HFAPI SPECT findings were compared with disease sites and the extent of HRCT. IPF patients exhibited elevated ^99m^Tc-HFAPI uptake in fibrotic abnormal regions, as shown by HRCT. A strong linear correlation was found between ^99m^Tc-HFAPI uptake and disease extent on HRCT. A moderate correlation was also found between ^99m^Tc-HFAPI uptake and PFT results. 

^18^F-FDG is the most widely used PET tracer in nuclear medicine. The value of ^18^F-FDG PET/CT in the assessment of disease severity, therapy efficacy, and prognosis in IPF patients has been widely studied [6,26]. High ^18^F-FDG SUV parameters were associated with poor pulmonary function and increased risk of mortality [27,28]. In addition, the early response to anti-fibrotic treatments has been evaluated. Although a reduction in pulmonary ^18^F-FDG uptake was observed in an IPF mouse model, the same result was not obtained in subsequent clinical studies in IPF patients 3 months after the initiation of anti-fibrotic therapy [29]. Other PET tracers, including CXCR4-targeted [30], SSTR-targeted [31], αvβ6 integrin-targeted [32] agents, have also been studied. The elevated signal in affected lung regions in IPF patients demonstrated a significant correlation with HRCT score and may identify disease severity and predict anti-fibrotic treatment outcome. In general, these tracers do not visualize fibroblast activation, which plays a principal role in driving lung fibrogenesis in IPF. 

Recently, the application of FAP-targeted tracers demonstrated promising results. Bergmann and Distler et al. found that ^68^Ga-FAPI-04 significantly accumulated in fibrotic areas of the lungs in SSc-related ILD, especially in patients with extensive disease, previous ILD progression, or high EUSTAR activity scores. Moreover, increased baseline uptake was associated with disease progression, and changes in the tracer uptake were correlated with response to the anti-fibrotic agent nintedanib [19]. Another study by Rohrich et al. demonstrated elevated ^68^Ga-FAPI uptake in fibrotic ILD lesions, which had a positive correlation with the CT-based fibrosis index [20]. Yang et al. reported that the total SUV of ^68^Ga-FAPI-04 PET was significantly related to lung function decline in ILD patients, and FAP expression level was closely correlated with the abundance of fibroblastic foci on biopsy sections from these patients [21]. A preclinical study by Rosenkrans et al. found that ^68^Ga-FAPI-46 had the potential to detect the early onset of fibrotic injury, where current diagnostic tools such as HRCT or spirometry provided inadequate diagnostic sensitivity [17]. 

Since ^99m^Tc is a more widely used radionuclide and has a lower radiation dose than PET tracers, ^99m^Tc-HFAPI was developed as an alternative choice. In this study, we validated the feasibility of using a ^99m^Tc-labeled FAPI tracer in the evaluation of IPF disease. IPF patients exhibited characteristic subpleural and peripheral ^99m^Tc-HFAPI uptake that corresponded to regions of fibrotic abnormalities on HRCT. The uptake of the tracer was compared with the disease extent shown on HRCT. We found positive and significant correlations between the total SUV_max_ or LBR and total HRCT score, for SUV_max_ (r = 0.7839, F = 54.41, *p* = 0.0094) and LBR (r = 0.7402, F = 56.33, *p* = 0.0092). 

This is consistent with the previous literature, which indicated that ^68^Ga-FAPI PET signal intensity correlated with the fibrosis index and disease extent determined by HRCT [19,20]. Of note, significantly higher SUV_max_ and LBR were observed in the higher HRCT stage (*p* < 0.001), except for HRCT stage 4 (more than 75% affected pulmonary tissue). We thought one explanation could be that massive vesicles found in the honeycombing and reticular opacity regions of HRCT stage 4 might reduce tracer signal intensity. The other explanation could be that ^99m^Tc-HFAPI specifically detects fibroblast activity. The FAPI signal might be weaker in the late phase where matrix deposition and tissue remodeling occur than in the activation phase during the pathophysiologic fibrotic process of IPF. Further investigations are needed to verify the accuracy of this interpretation. In addition, we found moderate but not significant correlations between ^99m^Tc-HFAPI uptake and PFT parameters. A correlation between FAPI uptake and PFT parameters was also observed which showed that ^68^Ga-FAPI uptake was significantly correlated with the corresponding FVC, DLCO results, and the changes in the previous 1 year [19]. The relationship between changes in FAPI uptake and lung function needs further verification.

There are several limitations in our study. First, we have preliminarily verified the feasibility of the application of ^99m^Tc-HFAPI in ILDs. The total number of patients was rather small. A larger cohort of IPF patients and other types of ILDs is necessary to validate our data and conclusions. Next, the changes in ^99m^Tc-HFAPI uptake and clinical outcomes have not been included in our preliminary research. Further follow-up study is warranted to assess the relationships between the changes in ^99m^Tc-HFAPI uptake, clinical features, and prognosis. Anti-fibrotic therapy response should also be evaluated in future studies. Another limitation is the lack of tissue biopsy. All patients recruited in our research have typical IPF. A confident diagnosis could be made according to clinical imaging criteria [4], making biopsy not necessary. Therefore, the immunohistochemical expression of FAP and its correlation with ^99m^Tc-HFAPI uptake could not be confirmed in our report.

## 4. Materials and Methods

### 4.1. Study Design and Patient Characteristics

This is a pilot study of calculating radiation dosimetry, distribution, and performance of a ^99m^Tc-HFAPI imaging probe in IPF patients. This study was approved by the institutional review board of Peking Union Medical College Hospital, Chinese Academy of Medical Sciences and Peking Union Medical College (Ethics committee approval No. ZS-3038). Written informed consent was obtained before patient participation. All procedures were in accordance with the ethical standards of the institutional and national research committees and with the 1964 Helsinki Declaration and its later amendments or comparable ethical standards. A total of eleven patients (nine males and two females, 62.0 ± 5.8 years of age) diagnosed with interstitial lung diseases were enrolled from December 2021 to March 2022 and imaged using serial scintigraphy and SPECT/CT. All patients were diagnosed with idiopathic pulmonary fibrosis by an experienced pulmonologist according to American Thoracic Society/European Thoracic Society guidelines [4] based on clinical–radiologic distinctive features. All these patients were defined with the usual interstitial pneumonia (UIP) pattern according to HRCT.

### 4.2. HRCT Imaging Protocol and Analysis

Non-contrast breath-hold high-resolution CT (HRCT) of the chest region was performed using a multi-slice CT scanner (Siemens Healthcare, Munich, Germany) with a slice thickness of 0.6 mm, at 120 kV tube voltage and 90 mAs current. The signs of pulmonary fibrosis were defined as reticular opacity, honeycombing, traction bronchiectasis, or lung architectural distortion, with absent or minor ground-glass opacity. In the areas of usual interstitial pneumonia (UIP) pattern on the images, the extent of affected pulmonary tissue was determined with visual evaluation using a 5-point stage (0, no involvement; 1, involving up to 25%, 2, involving from 26% to 50%; 3, involving from 51% to 75%; 4, involving from 76% to 100%; to the nearest 10%) at 5 levels (1, aortic arch; 2, tracheal bifurcation; 3, the origin of the apical segmental bronchus of the right lower lobe; 4, the entrance of the lower right pulmonary vein in the left atrium; 5, a level above top of the right hemidiaphragm) in the axial orientation according to the previous literature [33,34]. HRCT stage for each patient at 5 levels was recorded. In addition, the total HRCT score of each patient was also evaluated by summing up the estimated percentage of interstitial lung involvement using the ‘weighting’ factor to correct for differences in each lung zone volume at each HRCT level, based on methods reported previously [35,36]. The ratio of the weights for the five levels was 0.129 for level 1, 0.190 for level 2, 0.222 for level 3, 0.228 for level 4, and 0.230 for level 5. The percentage of abnormal lung at each level was multiplied by the corresponding ratio; the sum of the adjusted percentage of abnormal lung areas gave the total HRCT score of each patient.

### 4.3. Pulmonary Function Test

Pulmonary function tests (PFTs) were performed in all patients in a standard methodology according to ERS/ATS guidelines [37]. The following parameters were documented, including forced expiratory volume in the first second (FEV1), forced vital capacity (FVC), FEV1/FVC ratio, and diffusing capacity of the lung for carbon monoxide (DLCO), expressed as percentages of the predicted normal values except FEV1/FVC ratio. The composite physiologic index (CPI) was also calculated based on the previous literature using the following formula: CPI = 91.0 − (0.65 × percent predicted DLCO) − (0.53 × percent predicted FVC) + (0.34 × percent predicted FEV1) [38].

### 4.4. Preparation of ^99m^Tc-HFAPI

HFAPI was obtained by conjugating ^99m^Tc-chelator moiety (6-hydrazinonicotinamide, HYNIC) with the FAP targeting moiety (Appendix A). A single-dose lyophilized HFAPI kit containing 20 μg HFAPI, 6.5 mg tricine, 5 mg Tris (3-sulfonatophenyl) phosphine sodium salt (TPPTS), 12.7 mg succinic acid, and 38.5 mg disodium succinate was prepared. ^99m^Tc-HFAPI was prepared via the HFAPI kit. In short, one milliliter of Na^99m^TcO_4_ solution (~1110 MBq) was added to the HFAPI kit, and the reaction mixture was heated to 95 °C for 15~20 min. After cooling down to room temperature, ^99m^Tc-HFAPI was added. The radiochemical purity of ^99m^Tc-HFAPI was >95% analyzed by radio-ITLC. Subsequently, the ^99m^Tc-HFAPI was then formulated in phosphate-buffered saline and passed through a 0.22 μm Millipore filter for imaging purposes immediately. The in vitro stability was evaluated with radio-HPLC.

### 4.5. Whole-Body Scintigraphy and SPECT/CT Imaging Protocol

No fasting, special diet, hydration, or other specific preparation was required on the day of the ^99m^Tc-HFAPI injection. ^99m^Tc-HFAPI was injected intravenously with a quick bolus injection in a volume up to 1 mL with a mean administered activity of 813.4 ± 57.3 MBq (range 695.6–888.0 MBq). Serial whole-body scintigraphy at 5 min, 20 min, 40 min, 1 h, 2 h, 3 h, 4 h, and 6 h post-injection was performed using a Philips Precedence scanner (Philips Healthcare, Amsterdam, Netherlands) equipped with a low-energy general-purpose collimator, 20% symmetric energy window, with a peak at 140 keV photopeaks of ^99m^Tc. The whole-body anterior and posterior scintigraphy was acquired with the camera configured for dual-head planar imaging with a 256 × 1024-pixel matrix at a scan speed of 18 cm/min. Based on biodistribution results, ^99m^Tc-HFAPI SPECT/CT imaging of the chest region was performed after 4 h whole-body scintigraphy, with 32 frames and a 40 s exposure time per frame for each tomographic scan, including the chest region, followed by a low-dose CT acquisition (for attenuation correction and anatomic localization). The SPECT data were normalized and corrected for attenuation, decay, and scatter with the manufacturer-supplied reconstruction technique using an iterative ordered-subset maximum-likelihood expectation maximization algorithm with 3 iterations and 8 subsets. 

Vital signs were measured, a physical examination was performed, and any possible adverse clinical symptoms were documented within 1 h before ^99m^Tc-HFAPI injection, during the scans, and at 24 h after injection.

### 4.6. Distribution and Dosimetry Calculation

Visual analysis was used to determine the general distribution and the temporal and inter-subject stability. The dosimetry calculation was performed according to the European Association of Nuclear Medicine Dosimetry Guidance [39] and was conducted in seven representative patients out of a total of eleven patients. The target organs, including lungs, liver, spleen, kidneys, thyroid, small intestine, large intestine, and urinary bladder, and the remainder of the body were manually drawn on whole-body planar images. The activities were extracted from the regions of interest (ROIs) using the geometric mean of both the anterior and posterior projections of the planar images. The number of disintegrations for the target organs was obtained by fitting the data using a mono-exponential or a bi-exponential model, and the integration of the time–activity curve was calculated for every target organ. The curve-fitting and effective radiation dose calculation was performed using HERMES Hybrid Viewer 4.0 Dosimetry (HERMES Medical Solutions) according to the OLINDA/EXM (version 2.0) methodology. The absorbed dose of different organs and effective dose were calculated for dosimetry estimation as well. In addition, SPECT images were visually evaluated for the presence of FAP-positive lung parenchyma areas. For quantitative or semi-quantitative analysis, ROIs were drawn on transaxial images in the affected fibrotic lung region with increased tracer uptake on fused SPECT/CT images. The standardized uptake value (SUV) was calculated using the patient weight, injected activity, and ^99m^Tc camera calibration factor with HERMES HybridRecon standardized uptake value (SUV) SPECT (HERMES Medical Solutions) [40]. We recorded SUV_max_ from the highest voxel value in the fibrotic region of the image slice. The lesion-to-background ratio (LBR), namely the SUV_max_ of affected lung parenchyma divided by SUV of the blood pool (measured by drawing a 15 mm circle in the aorta arch), was calculated to adjust the differences among individuals. The SUV_max_ of FAP-positive affected lung parenchyma and LBR at 5 sections of each lung, corresponding to HRCT, were recorded. The SUV_max_ and LBR at each lung section were also summed up into a total SUV_max_ and LBR in each patient.

### 4.7. Statistical Analysis 

All statistical analysis was performed using GraphPad Prism software, version 6.0. (GraphPad Software Inc., San Diego, CA, USA). ^99m^Tc-HFAPI findings were compared with HRCT results. The SUV_max_ and LBR were plotted by different HRCT stages. Correlation was performed between the ^99m^Tc-HFAPI results and total HRCT score or PFTs values. Correlations analyses were assessed using the Pearson correlation test. Continuous variables were compared using an independent sample one-way ANOVA test. For all tests, a *p*-value < 0.05 was considered statistically significant. All quantitative data were expressed as mean ± standard deviation (SD).

## 5. Conclusions

Our initial findings demonstrated that ^99m^Tc-HFAPI SPECT is a promising new imaging modality in IPF patients. A dosimetry study showed the effective dose of ^99m^Tc-HFAPI was comparable to those of known ^99m^Tc tracers and was lower than those of the ^68^Ga- and ^18^F-labeled FAPI tracers. Further investigations with larger patient cohorts are warranted to evaluate the potential clinical value of ^99m^Tc-HFAPI SPECT imaging.

## Figures and Tables

**Figure 1 pharmaceuticals-16-01434-f001:**
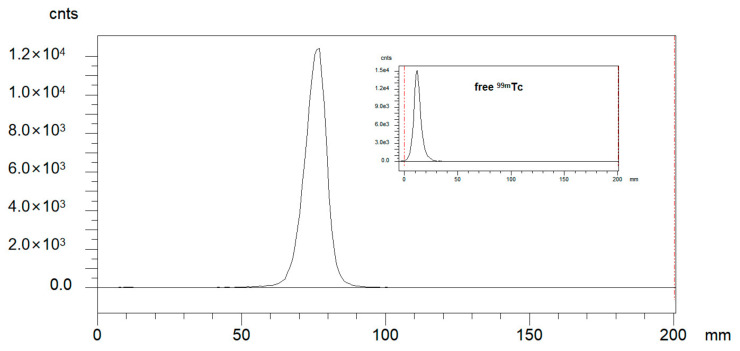
Integration diagram of analytical radio-iTLC radioactive traces of ^99m^Tc-HFAPI and free ^99m^Tc.

**Figure 2 pharmaceuticals-16-01434-f002:**
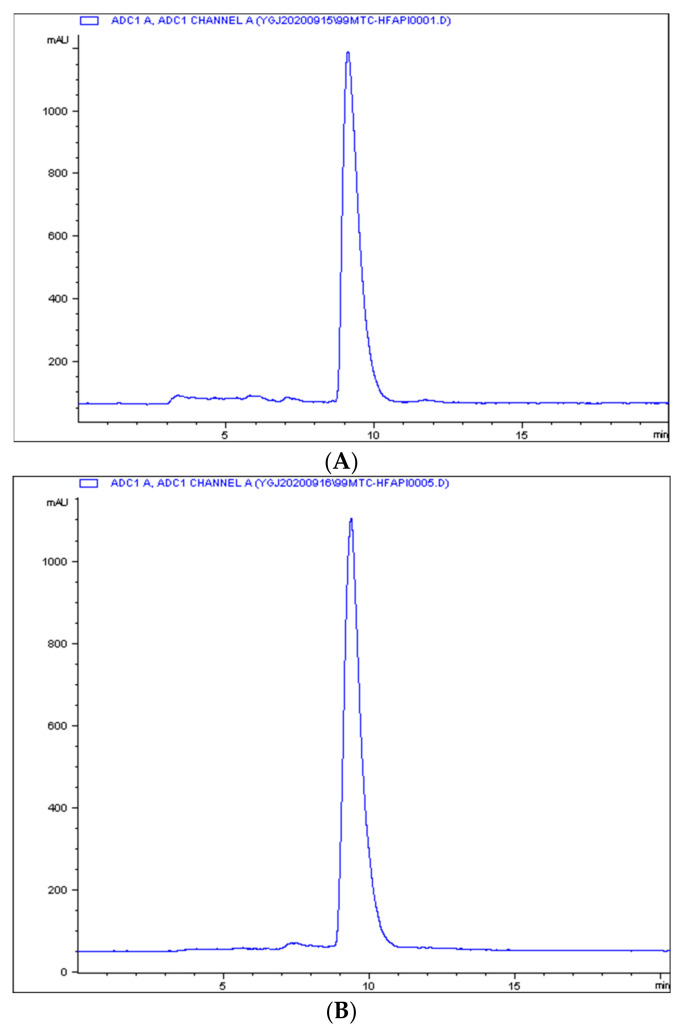
The in vitro stability of ^99m^Tc-HFAPI. Integration diagram of analytical radio-HPLC of ^99m^Tc-HFAPI at labeling (**A**) and 2 h in cystine (**B**).

**Figure 3 pharmaceuticals-16-01434-f003:**
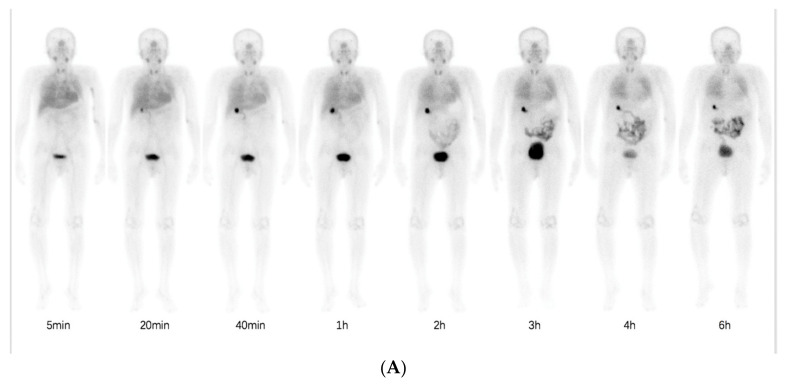
Representative whole-body anterior projection images of a 59-year-old male patient at different time points (5 min, 20 min, 40 min, 1 h, 2 h, 3 h, 4 h, and 6 h) after intravenous injection of ^99m^Tc-HFAPI (**A**). The semi-quantification of signal intensity ratios over time using whole-body planar images. The signal intensity ratios are the ratios of geometric mean counts of lung to whole body from anterior and posterior counts (**B**).

**Figure 4 pharmaceuticals-16-01434-f004:**
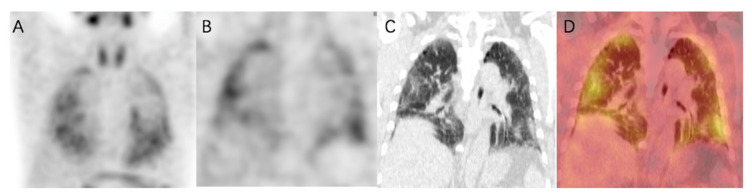
^99m^Tc-HFAPI SPECT/CT maximum-intensity-projection (MIP) (**A**), coronal SPECT (**B**), CT (**C**), and fused (**D**) images of a 65-year-old IPF patient. Increased tracer uptake is observed in a subpleural and peripheral distribution, which matches the fibrotic abnormalities in CT images.

**Figure 5 pharmaceuticals-16-01434-f005:**
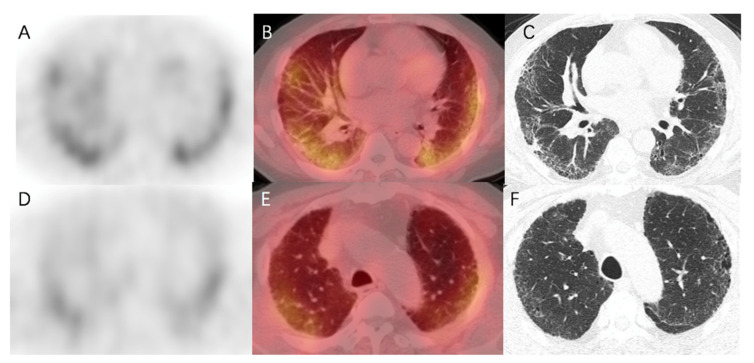
^99m^Tc-HFAPI SPECT/CT and HRCT images of a 65-year-old IPF patient. High ^99m^Tc-HFAPI uptake (SUV_max_ of 4.5 and LBR of 3.0 in the right lung and SUV_max_ of 4.4 and LBR of 2.9 in the left lung) in the subpleural of basal parts corresponded to the areas of reticular opacity on HRCT images (**A**–**C**). Minor abnormality in HRCT images in the apex parts level demonstrated an SUV_max_ of 3.4 and LBR of 2.3 in the right lung and SUV_max_ of 3.7 and LBR of 2.5 in the left lung (**D**–**F**).

**Figure 6 pharmaceuticals-16-01434-f006:**
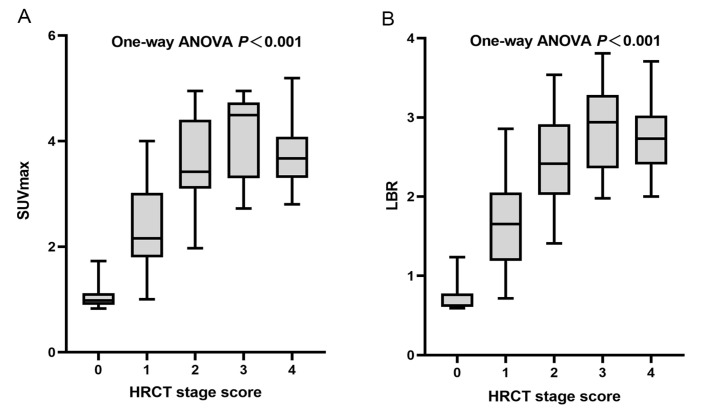
Plots of SUV_max_ and LBR according to HRCT stage. (**A**) Mean (SD) SUV_max_ were 1.09 (0.30), 2.37 (0.77), 3.68 (0.87), 4.48 (1.16), and 3.77 (0.64) for stage 0, 1, 2, 3, and 4, respectively (one-way ANOVA, *p* < 0.001). (**B**) Mean (SD) LBRs were 0.74 (0.22), 1.64 (0.55), 2.47 (0.53), 2.87 (0.55), and 2.72 (0.45) for stage 0, 1, 2, 3, and 4, respectively (one-way ANOVA, *p* < 0.001).

**Figure 7 pharmaceuticals-16-01434-f007:**
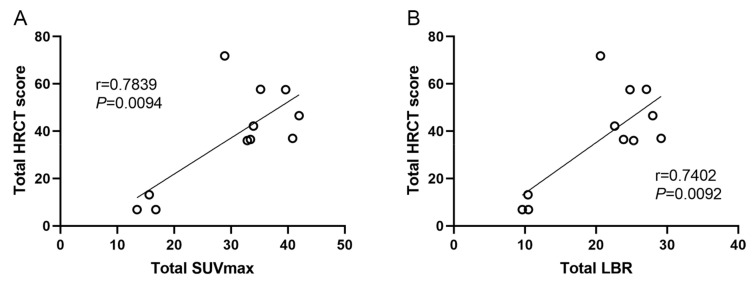
Linear correlation between total SUV_max_ or LBR with total HRCT score. (**A**) Total SUV_max_, r = 0.7839, *p* = 0.0094. (**B**) Total LBR, r = 0.7402, *p* = 0.0092.

**Table 1 pharmaceuticals-16-01434-t001:** Baseline demographic data for the study patients.

Baseline Demographic Data	N = 11
Age mean (range)	62 (55–75)
Gender	
Male	9
Female	2
Smoking Status	
Ex-smoker/Never-smoker	9/2
Pulmonary function tests, mean (range; SD)	
FVC%	69.4 (71.8–83.8; 23.2)
FEV1%	73.8 (75.3–91.5; 25.0)
FEV1/FVC%	97.4 (102.3–109.1; 30.4)
TLC%	64.4 (67.3–78.8; 20.7)
DLCO%	39.1 (27.8–51.0; 15.7)
CPI, mean (range; SD)	45.7 (40.8–61.5; 13.7)
Total HRCT score, mean (range; SD)	36.7 (24.6–52.1; 21.4)
SUV_max_, mean (range; SD)	4.1 (2.1–5.2; 1.0)
LBR, mean (range; SD)	2.1 (0.6–3.8; 0.9)
Total SUV_max_, mean (range; SD)	30.2 (13.5–42.0; 10.4)
Total LBR, mean (range; SD)	21.1 (9.6–29.2; 7.4)

Definition of abbreviations: N, number; FEV1, forced expiratory volume in the first second; FVC, forced vital capacity; DLCO, diffusion capacity for carbon monoxide; CPI, composite physiologic index; SD, standard deviation; SUV, standardized uptake value; LBR, lesion-to-blood-pool ratio.

**Table 2 pharmaceuticals-16-01434-t002:** Organ effective doses of ^99m^Tc-HFAPI.

ICRP-103 Effective Doses (mSv/MBq)		
Target Organs	Mean	SD
Adrenals	3.93 × 10^−5^	4.82 × 10^−5^
Gallbladder Wall	1.09 × 10^−4^	5.61 × 10^−5^
Colon	5.86 × 10^−4^	9.06 × 10^−5^
Small Intestine	5.35 × 10^−5^	2.23 × 10^−5^
Stomach Wall	3.34 × 10^−4^	2.38 × 10^−5^
Heart Wall	2.69 × 10^−5^	2.39 × 10^−5^
Kidneys	9.19 × 10^−5^	2.16 × 10^−5^
Liver	1.76 × 10^−4^	5.29 × 10^−5^
Lungs	7.16 × 10^−4^	1.54 × 10^−4^
Pancreas	3.37 × 10^−5^	4.87 × 10^−5^
Spleen	3.24 × 10^−5^	9.13 × 10^−5^
Thyroid	7.08 × 10^−4^	1.70 × 10^−4^
Urinary Bladder Wall	4.88 × 10^−4^	1.97 × 10^−4^
Prostate	1.56 × 10^−5^	2.74 × 10^−5^
Testes	7.09 × 10^−5^	5.95 × 10^−5^
Breasts	1.79 × 10^−4^	4.95 × 10^−5^
Ovaries	1.35 × 10^−4^	1.06 × 10^−5^
Uterus	1.77 × 10^−5^	2.19 × 10^−6^
Total body	4.11 × 10^−3^	5.59 × 10^−4^

## Data Availability

The data presented in this study are available on request from the corresponding author.

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
