# Peer review of "99mTc-Labeled FAPI SPECT Imaging in Idiopathic Pulmonary Fibrosis: Preliminary Results"

_pharmaceuticals, 2023, doi:10.3390/ph16101434_

Round 1
Reviewer 1 Report
This is a small pilot study of the potential utility of Tc-99m FAPI imaging in patients with idiopathic pulmonary fibrosis. The authors wish to extend further preliminary evidence which used Ga-68 FAPI in order to make the technique more widely available and less expensive. The authors found a positive relationship between FAPI avidity and disease stage determined by hi res CT, and a moderate correlation between FAPI avidity and hi res CT score for individual patients.
MINOR
Page 4, line 172. It is stated that ~740 MBq pertechnetate was added to the kit. However, after incubation, cooling, reformulation, and filtration, the mean administered activity was 813 MBq. Therefore, a significantly higher activity than 740 MBq must have been added to the kit.
Page 4, line 174. Radio-ITLC allows determination of radiochemical purity, not radiochemical yield
Page 10, line 317. It is stated that the mean whole-body effective dose was 0.0041 mSv/patient. However, it is actually 0.0041 mSv/MBq or 3.33 mSv/patient receiving 813 MBq of FAPI. However, the conclusion that this is comparable to other tracers is correct
TYPOS ETC
Page 2, Introduction, lines 87-88. Suggest changing wording of “Currently, the evidences of FAPI tracers’ application in ILDs are not many by far.” with “There have been few reports of application of FAPI tracers in ILDs.”
Page 2, line 93. Remove sentence which is essentially a repeat of the above sentence
Figure S2. Should say “vortex” rather than “vertex”
The references are not formatted consistently and are the isotope mass numbers should be superscripts rather than in parentheses
References 18 & 41. Page numbers missing (though ref 18 may still be in press)
The manuscript will require some editing for English grammar and idiom
Will need some editing for English grammar and idiom
Author Response
Dear Reviewer,
Thank you for your letter and for the comments concerning our manuscript entitled "99mTc-Labeled FAPI SPECT Imaging in Idiopathic Pulmonary Fibrosis: Preliminary Results" (Manuscript ID: pharmaceuticals-2535732). We greatly accept your valuable suggestions which might be of great help to improve the quality of our manuscript, as well as the important guiding significance to our researches. We have revised the manuscript, according to the comments and suggestions of reviewer, and responded, point by point to the comments as listed below.
- Response to comment:Page 4, line 172. It is stated that ~740 MBq pertechnetate was added to the kit. However, after incubation, cooling, reformulation, and filtration, the mean administered activity was 813 MBq. Therefore, a significantly higher activity than 740 MBq must have been added to the kit.
Response: Thanks for your notification, and we are very sorry we have made a mistake here, the activity that added to the kit should be 1110 MBq (corrected in the manuscript).
- Response to comment:Page 4, line 174. Radio-ITLC allows determination of radiochemical purity, not radiochemical yield
Response: We are very sorry we have made a wrong usage here, and it has been corrected in the revised manuscript.
- Response to comment:Page 10, line 317. It is stated that the mean whole-body effective dose was 0.0041 mSv/patient. However, it is actually 0.0041 mSv/MBq or 3.33 mSv/patient receiving 813 MBq of FAPI. However, the conclusion that this is comparable to other tracers is correct.
Response: We are very sorry we have used a wrong unit here, it has been corrected in the revised manuscript.
- Response to comment:Page 2, Introduction, lines 87-88. Suggest changing wording of “Currently, the evidences of FAPI tracers’ application in ILDs are not many by far.” with “There have been few reports of application of FAPI tracers in ILDs.”
Page 2, line 93. Remove sentence which is essentially a repeat of the above sentence
Figure S2. Should say “vortex” rather than “vertex”
The references are not formatted consistently and are the isotope mass numbers should be superscripts rather than in parentheses
References 18 & 41. Page numbers missing (though ref 18 may still be in press)
The manuscript will require some editing for English grammar and idiom.
Response: Response: Thanks for your kind suggestions. All the improper usages have been revised and the format of references has been standardized.
The English grammar and idiom have been edited.
We tried our best to improve the manuscript and made some changes. These changes will not influence the content of the paper. And the changes we did not list here were marked in red in revised manuscript.
We appreciate for your warm work earnestly, and hope the correction will meet with approve.
Once again, thank you very much for your comments and suggestions.
Best regards,
Yu Liu and Yitian Wu
Reviewer 2 Report
Interesting paper dealing with the application of Tc-FAPI in IPF. There is a renewed interest in Tc- radiopharmaceutical after the introduction of CZT solid state detector, therefore Tc-FAPI could be an emerging one.
However two points need to be elucidated and discussed:
*1 Preparation of 99mTc-HFAPI The authors claim a labelling yield of >95%, however the in vivo stability seems lower. In fact (fig 1, 2A and S2) there is an appreciable uptake in thyroid and salivary gland indicative of free technetium, as early as 5 minutes p.i. This impurity does not seem able to influence the pulmonary uptake but might introduce errors in the dosimetry evaluation. In which way the QC (TLC saline/MEK solvents ?) was performed?, other technical approaches were employed , for instance HPLC ?. In vitro stability (as for instance in serum /DTPA challenge), has been evaluated ?
*2 Serial scans were taken from 5 min pi up to 6 hrs, however it is not clear at which time the images, used for clinical analysis and correlation with the other parameters, were taken. Are they the 4 hrs SPECT images? Rohrich et al (ref.20) employing Ga-FAPI reported a different activity course in ILD and tumors. Are the time activity courses obtained with Tc-FAPI superimposable to those observed with 68 Ga, as far as SUV, Lesion to Bkg ratio and time to peak are concerned? Taking into account the longer half-life of Tc in comparison to the 68 min of Ga, the kinetics of Tc-FAPI could offer better insight in the disease activity
Reviewer 3 Report
1-Line 41 in abstract, please revise ‘’significant’’ to ‘’significantly’’
2-Line 244, Revise ‘’Drug-related’’ to ‘’toxic-related’’.
3- Could you please provide information about in vitro stability of 99mTc-HFAPI tracer?
4-Line 359-360, please revise ‘’significant’’ to ‘’significantly’’

Round 2
Reviewer 2 Report
Your answer and comments are satisfactory, even if I have some doubts about “the physiological uptake” in thyroid and salivary glands which is not observed with Ga-FAPi-
Anyway I do not understand the reason why the exhaustive data, you furnished in the letter, were not included in the text. I think that they are worthwhile and can improve the paper quality and interest.
For this reason I left unchanged the “must be improved” assessment about methods and results.
